# Biofeedback Respiratory Rehabilitation Training System Based on Virtual Reality Technology

**DOI:** 10.3390/s23229025

**Published:** 2023-11-07

**Authors:** Lijuan Shi, Feng Liu, Yuan Liu, Runmin Wang, Jing Zhang, Zisong Zhao, Jian Zhao

**Affiliations:** 1College of Electronic Information Engineering, Changchun University, Changchun 130022, China; 2Jilin Provincial Key Laboratory of Human Health Status Identification Function & Enhancement, Changchun 130022, China; 3Key Laboratory of Intelligent Rehabilitation and Barrier-Free for the Disabled, Changchun University, Ministry of Education, Changchun 130012, China; 4College of Cyber Security, Changchun University, Changchun 130022, China; 5College of Computer Science and Technology, Changchun University, Changchun 130022, China

**Keywords:** virtual reality, biofeedback, breathing interaction, rehabilitation training

## Abstract

Traditional respiratory rehabilitation training fails to achieve visualization and quantification of respiratory data in improving problems such as decreased lung function and dyspnea in people with respiratory disorders, and the respiratory rehabilitation training process is simple and boring. Therefore, this article designs a biofeedback respiratory rehabilitation training system based on virtual reality technology. It collects respiratory data through a respiratory sensor and preprocesses it. At the same time, it combines the biofeedback respiratory rehabilitation training virtual scene to realize the interaction between respiratory data and virtual scenes. This drives changes in the virtual scene, and finally the respiratory data are fed back to the patient in a visual form to evaluate the improvement of the patient’s lung function. This paper conducted an experiment with 10 participants to evaluate the system from two aspects: training effectiveness and user experience. The results show that this system has significantly improved the patient’s lung function. Compared with traditional training methods, the respiratory data are quantified and visualized, the rehabilitation training effect is better, and the training process is more active and interesting.

## 1. Introduction

With the rise of modern industrialized cities, fossil fuel emissions have also increased air pollution, seriously endangering the health of the respiratory system [1]. In China, the number of patients with chronic obstructive pulmonary disease (COPD) has exceeded 100 million, and is still on the rise, according to the “China Adult Lung Health Study” released in 2022. According to the World Health Organization, COPD is the third leading cause of death in the world [2]. So far, the global prevalence and mortality rates have further increased [3], seriously affecting patients’ daily travel problems [4], which shows that it is extremely harmful. As the number of COPD patients increases year by year, research on the prevention of respiratory diseases is particularly important.

The improvement of people’s awareness of prevention has led to a greater development of preventive science and rehabilitation, among which respiratory rehabilitation treatment is a branch of rehabilitation medicine. Respiratory rehabilitation application [5] refers to the use of active and effective respiratory exercise training to reduce the symptoms of patients’ dyspnea and improve the patient’s lung function, thereby improving the patient’s quality of life. Therefore, rehabilitation training for respiratory diseases is very necessary.

Conventional respiratory function training includes cough training, pursed-lip breathing, abdominal breathing, balloon breathing training, etc. [6]. Xiaoyan Liu [7] and others conducted pulmonary function training on patients with COPD through pursed-lip breathing and abdominal breathing training, which effectively improved the patients’ pulmonary function. Liu Guofang [8] used abdominal breathing training combined with balloon training in the treatment of patients with chronic persistent asthma, and Zheng Liping [9] used intensive cough breathing training for patients undergoing pulmonary drainage surgery, which significantly improved lung function and various physical conditions. Noman Sadiq [10] and others studied the problem of breathing exercise regulating heart rate variability and found that the breathing exercise of blowing a balloon enhanced lung function.

Through the analysis of the above studies, it is found that regular breathing training can improve the lung function and symptoms of dyspnea in people with breathing disorders. However, the breathing training process is monotonous and boring, and there are problems such as unquantified breathing data and poor feedback of breathing training results. The effect of the patient’s breathing training and the real-time breathing training data cannot be reflected in time.

With the development of virtual reality technology, people have introduced it into rehabilitation training. This study applies virtual reality technology combined with biofeedback to respiratory rehabilitation training to visualize respiratory training data and intuitively understand the effect of respiratory rehabilitation training. This system can not only improve users’ attention and enthusiasm for training, but also enable users to train efficiently and orderly through personal training tasks.

## 2. Related Work

### 2.1. Application of Virtual Reality Technology in Medical Field

With the development of virtual reality technology, virtual reality technology has penetrated into many medical fields. In the rehabilitation training of neuromotor injuries [11], the use of virtual reality scenes can provide patients with sensory experience and play an important role in their rehabilitation training. In psychotherapy [12], the use of virtual reality technology for mindfulness intervention can distract attention and reduce anxiety and stress. In respiratory rehabilitation training, KC Lan [13] studied multi-modal virtual reality for respiratory training, which improved the enthusiasm of patients for respiratory rehabilitation training. Colombo [14] and others summarized and analyzed the latest technology in the rehabilitation of COPD patients using virtual reality technology, which can distract patients from fatigue and dyspnea. Mitsea et al. [15] introduced a virtual slow breathing training system that provides immersive relaxation scenes to achieve a stable breathing rate. Irini Patsaki [16] applied VR rehabilitation training in the treatment of COPD patients to effectively increase the treatment effect. In rehabilitation training after musculoskeletal injuries [17], scholars have developed virtual reality-based rehabilitation training to help athletes recover from musculoskeletal injuries. In low vision rehabilitation [18], researchers summarized the application of virtual reality technology in visual field expansion and visual acuity improvement in the rehabilitation of low vision people, and found that virtual reality equipment can expand the visual field and improve visual acuity. In the rehabilitation training of children with autism [19], virtual reality technology has good effects in training language skills, life skills, and emotional expression. In the rehabilitation treatment of stroke patients [20], the designed virtual scene is used for rehabilitation treatment, so that the patient’s neuromotor system begins to recover its functions.

In the above research, virtual reality technology can provide an immersive virtual scene, using the advantages of the virtual scene to make the originally boring content lively and interesting, and improve the enthusiasm and participation of participants. In the process of rehabilitation training, the patient’s attention is improved, mental stress is relieved, and the effect of rehabilitation training is improved.

### 2.2. Biofeedback Technology

Biofeedback is the breathing data detected by sensors and fed back to the user in real time. It can observe one’s own breathing data in real time and improve the user’s behavior through feedback [21]. Biofeedback is also a form of stress reduction, providing physical and psychological comfort. Sebastian Rutkowski et al. [22] have shown that training parameters that provide feedback will increase patients’ motivation to actively achieve health. Zhou Lina [23] and others studied that stress reduction therapy combined with music biofeedback therapy can reduce patients’ negative emotions and improve their sleep quality. The combination of virtual reality technology and biofeedback has shown better results. The virtual reality game designed by Rakesh Patibanda [24] and others is combined with biofeedback. Changes in breathing rhythm drive the color changes of trees and the lushness of branches, thereby giving feedback to the patient’s own breathing status. Robert Greinacher [25] and others have significant effects on adjusting the user’s breathing rhythm through tactile and visual feedback of virtual scenes. Johannes Blum [26] and others proved that in VR-based breathing training, the feedback method of environmental biological changes has lower cost and better effectiveness in increasing breathing awareness and promoting diaphragmatic breathing.

Through the analysis of the above studies, it is found that biofeedback technology has been widely used in the field of medical rehabilitation. It increases the patient’s awareness of breathing, helps them assess their current breathing pattern, can improve the patient’s breathing pattern if necessary, and provides patient learning through sensory feedback stimulation. When biofeedback is combined with virtual reality technology, the effect of feedback will be more obvious and vivid, which can improve the effect of self-feeling during rehabilitation training. But so far, during rehabilitation training, most patients with virtual reality need to wear heavy physiological sensors. For patients who are physically inconvenient or recovering in hospital beds, there will be movement restrictions and physical burdens, which prevent them from fully achieving a high degree of immersion and easy and natural interaction [27].

## 3. System Architecture

In order to overcome the above obstacles, a biofeedback respiratory training rehabilitation system based on virtual reality technology is designed to improve the patient’s respiratory function and achieve visual biofeedback to obtain the patient’s own real-time respiratory training effects and data. The system is designed to be an intuitive and portable breathing training system that does not require expensive and heavy measurement sensors. A portable vital capacity sensor is used combined with HTC Vive pro2 VR equipment to form a respiratory rehabilitation training system that uses respiratory data to drive virtual scene changes. The respiratory training effect is presented in the form of visual feedback in biofeedback. This paper provides an overview of the designed respiratory rehabilitation training system and conducts an empirical evaluation of its experience and training effects.

### 3.1. System Framework Structure

This system is a set of rehabilitation training system built to simulate the real environment for the rehabilitation needs of patients with respiratory disorders. The system uses the virtual reality scene developed by Unity 3D as the core, and builds a scene processing system. Respiratory data are used as the input part of the training system, and the head-mounted display is used as the output part of the system data. Patients can be immersed in the virtual environment for respiratory rehabilitation training. The respiration data are collected through the respiration sensor, and the respiration data are sent to the PC and interacts with the virtual scene through the interactive processing unit. The spatial locator recognizes the position of the person through the handle controller and the sensor in the head-mounted display, and maps the recognition result to the virtual training scene to obtain the spatial position in the current virtual scene. Finally, the virtual scene is output through the head-mounted display, and the respiratory data are presented to the patient in the form of visual feedback. Finally, wecomplete the task of system design. The system framework structure is shown in Figure 1.

### 3.2. Hardware Equipment

The VR device used in this study is HTC Vive pro2 device. HTC Vive pro2.0 equipment includes handle controller, head-mounted display, base station and HDMI equipment, as shown in Figure 2. The device is equipped with ultra-high resolution, allowing users to enjoy more delicate images in the virtual reality experience, while providing 3D spatial sound effects and an immersive sound experience, allowing users to better immerse themselves in virtual reality.

The HTC Vive pro2.0 device uses infrared tracking technology, which emits infrared rays from the base station transmitter and covers the entire room. Since the handle controller and the head-mounted display are equipped with a large number of sensors, the base station recognizes the sensors through infrared rays, so as to obtain the current external location of the handle controller and the head-mounted display. Data transmission and tracking technology are shown in Figure 3.

The respiratory data sensor used is the HKF-20C vital capacity sensor. The HKF-20C vital capacity sensor is a sensor used to measure vital capacity and respiratory function. It introduces the breathing gas of the measured person into the sensor through the breathing tube for analysis and calculation. The sensor will measure the flow rate, volume and time of the subject’s inhalation and exhalation, thereby calculating vital capacity, respiratory rate, breathing depth and other indicators. Figure 4 shows the HKF-20C vital capacity sensor.

The respiratory data sensor mainly collects respiratory signals and transmits the data to the PC in real time, which is used as a basis for judging the effect of rehabilitation training. The head-mounted display maps the virtual scene onto the display, providing patients with a realistic and immersive virtual environment. The hand controller serves as a controller in the virtual environment, performing functions such as selecting virtual scenes and moving virtual characters. The base station sets the size range of the real environment occupied by the virtual scene to standardize the movable range of breathing training. At the same time, the locator performs real-time data transmission with the head-mounted display and hand controller to track the user’s real-time location.

### 3.3. Interaction between Breathing Data and Virtual Scenes

Patients with respiratory disorders wear a respiratory data sensor and a head-mounted display to collect respiratory signals synchronously. At the same time, the respiratory data sensor will send the collected data to the PC, and then send the data to the training scene to interact with the training scene. Among them, various breathing training methods are built into the VR breathing training game scene. During the breathing interactive training process, patients can observe changes in the surrounding environment according to their own breathing rhythm. At the same time, the patient’s respiratory data will be detected, the respiratory information parameters will be calculated and the characteristic values of the respiratory airflow will be extracted to realize the visualization of the respiratory data. After the training is over, visual feedback is given to the patient in the form of quantification and visualization, which is conducive to discussing the shortcomings of their own breathing training, and improving it. The interaction between breathing data and virtual scenes is shown in Figure 5.

### 3.4. Virtual Scene Design

Playful feedback relieves patient fatigue during training and prevents overly tedious therapy, which is crucial for long-lasting rehabilitation. This article designs a simple and interesting biofeedback virtual environment to meet the needs of people with different degrees of respiratory impairment. It provides users with training conditions for deep breathing and shallow breathing. Through approachable and easy-to-operate training scenarios, it makes user training operations convenient and concise. When developing the system, this article designed two virtual scenes of “blowing candles” or “blowing dandelions”, as shown in Figure 6 and Figure 7. After the system completes “blowing candles” or “blowing dandelions”, the data after breathing training is presented to the patient in the form of visual feedback to evaluate the training effect of the patient.

The “blow out candles” scene is designed based on the indoor environment. The scene includes tables, chairs, corridors, halls, bathrooms, furniture and other items. The user performs breathing training according to the voice prompt instructions, walks to the candle and blows on the candle. When the maximum forced respiratory volume (FVC) reaches the set threshold, the candle is extinguished.

The “Dandelion Blowing” scene is designed with the theme of a park, which contains a variety of trees, buildings, vegetation, rockeries, etc. Users can walk freely in the park. There are many dandelions in the grass. According to the task prompt, the user picks up the dandelions for breathing training, and blows on the dandelions after picking them up. When FVC reaches the set threshold, the dandelions are blown away, and the trained data is visually output. The system sets that the user needs to complete five breathing exercises to complete the task. At the same time, it is equipped with a voice prompt function, which can increase the fun and experience of the user.

### 3.5. Breathing Data Collection Interaction and Visualization

#### 3.5.1. Respiratory Data Collection

The respiratory data are collected through the sensor, the data signal output of the vital capacity sensor is connected to the computer through the serial port, and the data are transmitted to the PC in the form of serial communication. First establish the communication format between the sensor and the scene-driven engine Unity3D, initialize the serial port, and configure the virtual serial port, baud rate, parity bit, data bit, and stop bit. Then, connect the disposable blowpipe to the sensor and hold the lower end of the sensor. After the preparation is completed, start to blow air. After blowing, data will be transmitted to the computer. When the breath stops or intermittent breath occurs during the blowing process, the sensor will determine that the data transmission has ended. At the same time, when collecting respiratory data, do not block the exhaust hole, otherwise it will affect the current vital capacity collection results. During use, avoid debris blocking the sensor’s internal detector, otherwise it will affect the accuracy of the sensor’s respiratory data collection.

#### 3.5.2. Analysis of Respiratory Data

Functional data parsing enables the study of complex data and can be applied to observations of functions that vary on a continuum that appear in reality [28]. In order to realize the visualization and digitization of data, it is necessary to analyze the respiratory signal data and extract its eigenvalues. First, after the respiratory data are uploaded, the sensor will detect the respiratory data, and then the computer will analyze the respiratory data according to the communication protocol, so as to obtain the waveform data during the sensor testing process. From the extracted waveform data, the characteristic points, peaks, etc. of the signal can be observed. At the same time, the signal characteristic values are extracted and stored, and then the measurement parameters, namely breathing time, vital capacity, flow rate, etc., are output at a sampling frequency of 100 HZ. After the parsing is completed, the data characteristic value is returned to the array for storage. Algorithm 1 shows the respiratory data parsing algorithm.

First, the respiratory data collected are cached in the cache as a byte array. If the cache contains data, the contained data are traversed. When the start measurement command 0xc1 of the respiration sensor and the end command 0xf0 of the respiration sensor are detected, the data in the cache cache are sequentially imported into endData for storage. Secondly, the byte type number endData is converted into a hexadecimal, and the collected data contain 25 bytes, as shown in Table 1. Among them, the first 21 bits obtain the high and low bytes of each type of breathing data, and the size of each measured data type is the combination of high and low bytes. Therefore, when i < 22, take an even number to combine the high and low bits. Then, use the ToString(“X2”) function to convert it to a hexadecimal, and finally output 16 for the lungData collection of breathing data.
**Algorithm 1:** Respiratory data analysis algorithm.Input: collected respiratory data collection “cache”Output: Hexadecimal respiratory data set “lungData”1. if cache.Count!=0 then2.      for int i = cache.Count-1; i >= 0; i– do3.        if i! = 0 then4.            if cache[i] == 0xc1&&cache[i-1] == 0xf0 then5.                Array.Copy(cache.ToArray(),i + 1,endData,0,endData.Length)6.                isReceived = true7.                cache.Clear()8.      if isReceived then9.          for int i = 0; i < endData.Length; i++ do10.              if i < 22 &&i % 2 == 0 then11.                  hex += endData[i].ToString(“X2”)12.                  hex += endData[i + 1].ToString(“X2”)13.                  lungData.data.Add(Explain(hex))14.              else if i < 25 then15.                  hex += endData[i].ToString(“X2”)16.                  data.Add(Explain(hex))17. lungData.GetDetail()

#### 3.5.3. Analysis of Respiratory Data

In the virtual scene, the trainee performs breathing training according to the instructions. When the FVC exceeds the set threshold (1000), the candle will go out. This is because the image waveform cannot intuitively and better reflect the size of the data change each time. Therefore, the data of each breathing training is analyzed. While the breathing data drives the scene changes, the analyzed data are called and the data are presented in a visual and digital form, so that the trainees can observe the data more clearly. Algorithm 2 shows the respiratory data interaction and visualization algorithm, and Figure 8 shows the visualization of respiratory data. The data types include the duration of insufflation (TIME), vital capacity (FVC), peak flow rate (FEF), flow rate at 25% vital capacity (MEF25), flow rate at 50% vital capacity (MEF50), and flow rate at 75% vital capacity (MEF75), flow velocity difference (FEF), 25–75% average flow velocity (PEF), vital capacity in the previous second (FEV1), vital capacity in the first two seconds (FEV2), vital capacity in the first three seconds (FEV3), vital capacity ratio in one second (V1F), vital capacity in two seconds (V2F) and vital capacity in three seconds (V3F).

First, assign values to the fields, fill the data data before assigning values, and store 14 data collected by the breath sensor into data. When the vital capacity is greater than the set threshold of 1000 mL (data.fvc > 1000), since the fire of the candle and the fluff of the dandelion are special effects, item.Stop() will turn off their special effects, and will use SetActive to turn off their active status. Then, determine whether the virtual target Target of the breathing training is the same as the original one. If the target changes, that is, the candle goes out or the dandelion fluff blows away, use StartPort() to visually output the blowing data locally, as shown in Figure 8.
**Algorithm 2:** Respiratory data interaction and visualization algorithm.1. if data.Count == 14 then2.      time = data[0],fvc = data[1],fef = data[2],mef25 = data[3],mef50 = data[4],         mef75 = data[5],fef25–75 = data[6],pef25-75 = data[7],fev1 = data[8],         fev2 = data[9],fev3 = data[10],v1f = data[11],v2f = data[12],v3f = data[13]3.       data.Clear()4. if data.fvc > 1000 then5.      foreach var item in particle then6.         item.Stop()7.         item.transform.GetChild(0).gameObject.SetActive(false)8. if Target ! = null then9.      if flag == false then10.         StartPort port = new StartPort()11.         byte[] data = GetData(port)12.         localClient.Send(data)13.         StartPort()14. DebugMessage.Log()15. Break

## 4. Experiments and Results

### 4.1. Participants

In order to understand the effectiveness of the system, this paper recruited 10 volunteers to participate in the test, and at the same time equipped two staff members to instruct how to use the research system; then, we trained the participants, distributed the equipment and managed the results of breathing training. The participants were divided into two groups: the first group was the control group, and 5 participants were given routine breathing training, blowing out candles and dandelion breathing training. The second observation group, the remaining 5 participants, performed breathing training in the virtual scene designed in this paper. The diagnosis of COPD was made according to a set of criteria developed by the Global Initiative for Chronic Obstructive Lung Disease (GPLD) [29]. To determine whether a person’s airway is blocked, clinicians compare the amount of air a patient can exhale in one second, known as forced expiratory volume (FEV1), to the total amount of air they can exhale, known as forced vital capacity (FVC). According to GOLD, a person can be diagnosed with COPD if their ratio of FEV1 to FVC is below 0.7, which means that a person exhales less than 70 percent of the air in their lungs in one second [30].

To ensure the validity of the assessment of effect, excluding patients with other lung function, participants met the following criteria:Meet the diagnostic criteria for COPD;During the rehabilitation training period, there is no resistance to cooperate with training and other behaviors;Good compliance during rehabilitation training and tolerance during training.

Table 2 shows the participants’ information. Participant screening is based on the fact that COPD patients are mainly concentrated in middle-aged and elderly people, and the age distribution is between 55 and 65 years old. At the same time, the prevalence of COPD in men is more than twice that of women in the same age group [31]. In order to eliminate the interference of participants with other medical conditions on the experiment, participants with no other medical conditions and who were able to perform breathing training on their own were selected for the experiment.

### 4.2. Experimental Process

After each breathing session, each participant will receive a questionnaire to fill out regarding the training experience. This scale examines participants’ feelings about training through the following questions:During training, do you feel bored?During the training process, are you distracted and have trouble concentrating?During the training process, are you able to keep up with the training pace?

Each item in this experimental survey adopts a five-point Likert scale, with options ranging from 1 to 5, that is, strongly agree to strongly disagree. Each item is scored from 1 to 5 points, and the score is multiplied by the corresponding weight to obtain the final score, as shown in Table 3.

### 4.3. Experimental Results and Analysis

During this respiratory rehabilitation training, the changes in respiratory data parameters of each participant before and after training were recorded to measure the effectiveness of the training system. The experiment found that the breathing training effect of each participant has improved. In this article, the breathing data of each participant before and after training are used as a measure of the effect of this experiment. At the same time, the mean values of the data before and after training for the two groups of participants were obtained for comparison.

Table 4 and Table 5 are the breathing data parameters before and after the rehabilitation training of the control group, and Table 6 and Table 7 are the breathing data parameters of the observation group before and after the rehabilitation training. Figure 9 and Figure 10 show the average breathing data of the control group before and after training, and Figure 11 and Figure 12 show the average breathing data of the observation group before and after training. Through Figure 9 and Figure 11, it can be found that the breathing data of the participants in the control group and the observation group have obvious changes before and after. Compared with the vital capacity and average flow rate in a short period of time, there are obvious improvements. The diagnosis of COPD patients is based on the ventilation volume in a short period of time, that is, the proportion of vital capacity in one second. From Figure 10 and Figure 12, it can be observed that the proportion of the vital capacity of the participants in the first second after training was significantly improved. It shows that both the traditional breathing training and the research system have significantly improved the respiratory function of the lungs of COPD patients. However, the flow rate (MEF75) at 75% vital capacity of the two groups of participants decreased to varying degrees. From Table 4, Table 5, Table 6 and Table 7, it can be observed that the MEF75 of individual participants decreased to varying degrees, because the sensor collected breathing data. The participants have insufficient strength, and the equipment produces corresponding noise and heat loss. In the later research, the respiratory data collection process and rehabilitation training process will be improved and optimized to improve the accuracy of the data.

It is observed in Table 4 and Table 5 that the vital capacity in the first second of all patients is low, but the vital capacity in the second and third seconds is high. This is because diseases such as pulmonary obstruction, emphysema, asthma, etc., can cause airway obstruction, making the lungs unable to exhale the full lung capacity in the first second. However, over time, lung capacity gradually increases in the second and third seconds, and the patient tries to expel as much gas as possible by increasing their breathing rate or changing their breathing pattern.

By observing Figure 13 and Figure 14, it is found that the training effect of the control group is significantly lower than that of the observation group. At the same time, the first-second vital capacity of the observation group increased by 5% compared with that before training, while the first-second vital capacity of the control group increased by 2% compared with that before training. It can be observed that the respiratory rehabilitation training of this research system is superior to the traditional respiratory rehabilitation training.

This experiment statistically summarizes the feelings of each participant during the training process, as shown in Figure 15, Figure 16 and Figure 17. From the figure, it can be observed that the observation group had high scores for feeling bored and bored after the first week of training, and low scores for inattention and ability to keep up with the rhythm, which may be caused by unfamiliarity with the system for the first time. Over the next few weeks, as the user mastered the use of the system, the scores for feelings of boredom also decreased, while the scores for feelings of lack of concentration and failure to keep up with the rhythm also gradually increased. In the control group, after the first week of training, the feeling of boredom was relatively low, and the scores of inattention and inability to keep up with the rhythm were higher. As the training progressed, the feeling of boredom increased during training, resulting in inattention and so on. This leads to an increase in the feeling score of boredom and a gradual decrease in the feeling score of the lack of concentration. On the whole, as the experiment progresses, this research system has a better sense of experience and provides users with a higher sense of immersion, participation and fun.

## 5. Conclusions

Traditional rehabilitation training lacks clear breathing data, and the training process is very tedious. In contrast, this article studies a biofeedback respiratory rehabilitation training system based on virtual reality technology, which reflects the interaction between respiratory data and virtual scenes during the rehabilitation training process, and realizes the quantification and visualization of respiratory data. At the same time, the effect of the user’s rehabilitation training is improved, and the enthusiasm and interest of the user are increased.

The purpose of this research system is to improve the lung function of patients with respiratory disorders and relieve symptoms such as dyspnea. Through VR equipment and respiratory data sensors, the interactive virtual scene of human breathing in the form of biofeedback is realized. We perform data analysis on the respiratory data, use the respiratory data to drive changes in the virtual scene, and realize the interaction between the respiratory data and the virtual scene. We use data visualization algorithms to quantify and visualize respiratory data to evaluate the effectiveness of patient breathing training. In the virtual scene of breathing training, interesting ways such as music and stories are used to guide patients to continuously perform breathing training in a comfortable process, so as to complete specific rehabilitation training tasks and goals. The system is evaluated from two aspects, including training effectiveness and user experience. Comparing the experimental results and experiences of 10 participants, the results show that compared with traditional respiratory rehabilitation training, this research system has better training effects and experience, and rehabilitation training is more positive.

The system enables patients with respiratory disorders to improve their respiratory function by completing training tasks in a virtual scene. It is easy to operate and has no restrictions on site and usage time, making it possible to incorporate it into a long-term pulmonary rehabilitation plan. Due to the interference of respiratory equipment in the process of respiratory data transmission, certain Gaussian white noise and random noise will be generated. The focus of follow-up research is to further improve and perfect the respiratory data collection and improve the accuracy of respiratory data transmission.

## Figures and Tables

**Figure 1 sensors-23-09025-f001:**
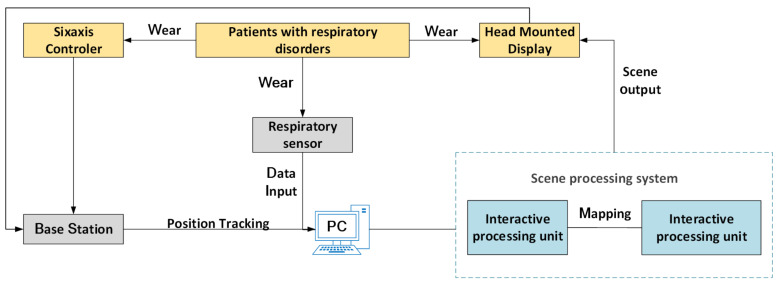
System frame structure diagram.

**Figure 2 sensors-23-09025-f002:**
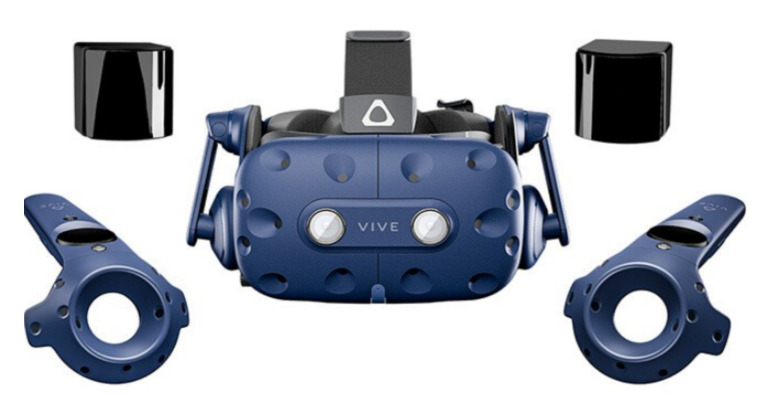
HTC Vive pro2 device.

**Figure 3 sensors-23-09025-f003:**
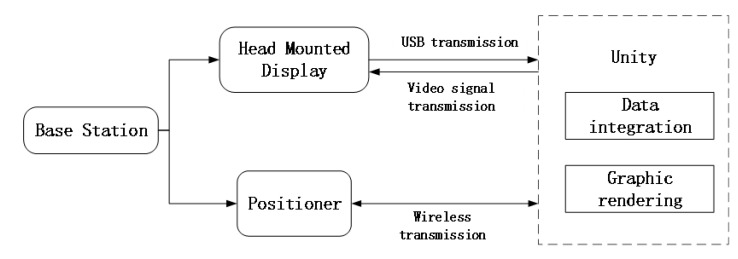
Data transmission and tracking technology.

**Figure 4 sensors-23-09025-f004:**
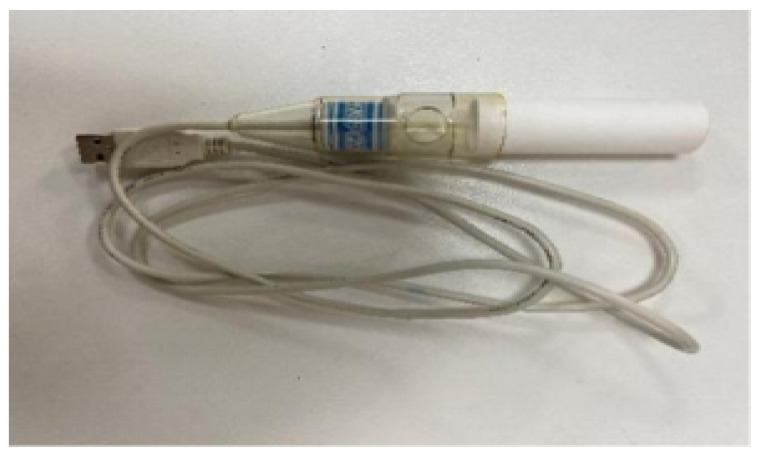
HKF-20C vital capacity sensor.

**Figure 5 sensors-23-09025-f005:**
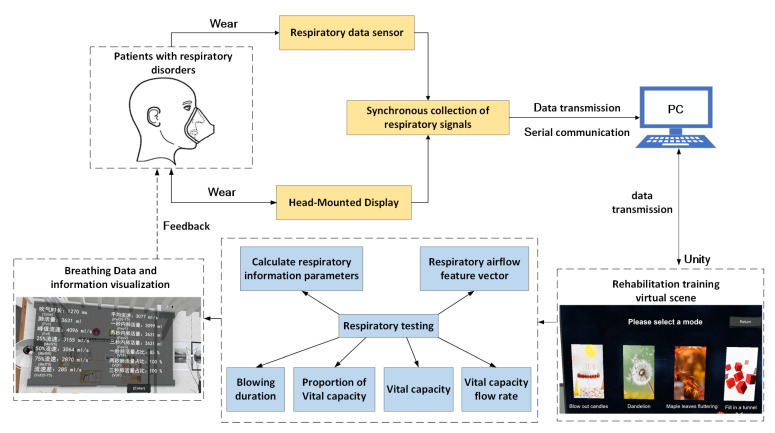
Interaction between respiratory data and virtual scene.

**Figure 6 sensors-23-09025-f006:**
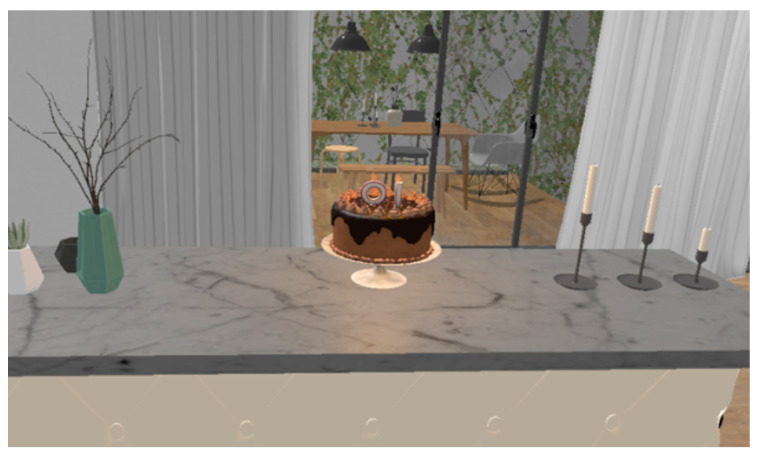
Blowing out the candle scene.

**Figure 7 sensors-23-09025-f007:**
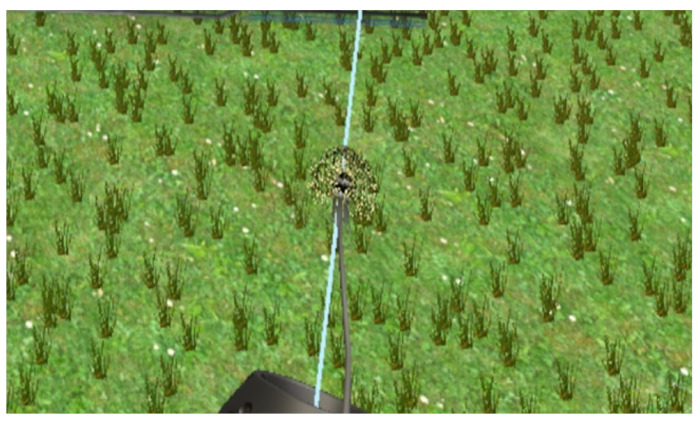
Blowing dandelion scene.

**Figure 8 sensors-23-09025-f008:**
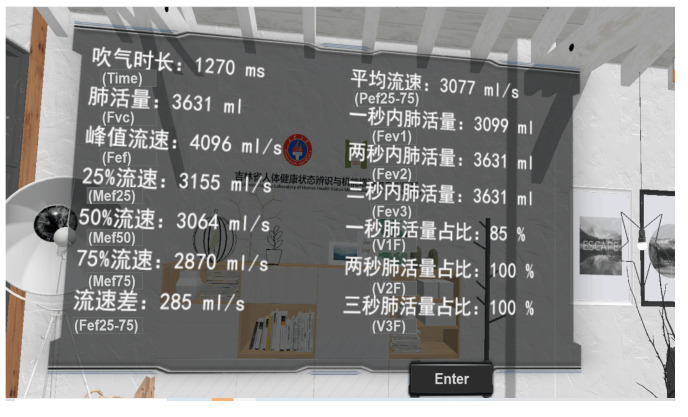
Respiration data visualization.

**Figure 9 sensors-23-09025-f009:**
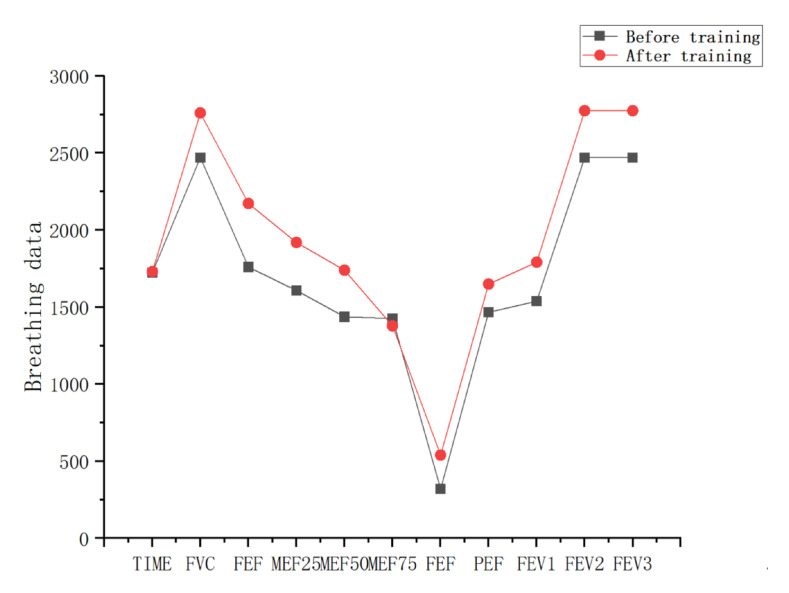
Average respiratory data before and after training in the control group.

**Figure 10 sensors-23-09025-f010:**
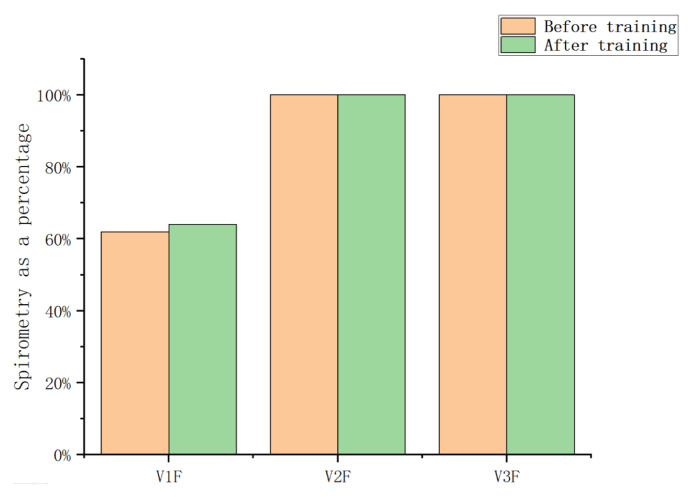
Average vital capacity ratio before and after training in the control group.

**Figure 11 sensors-23-09025-f011:**
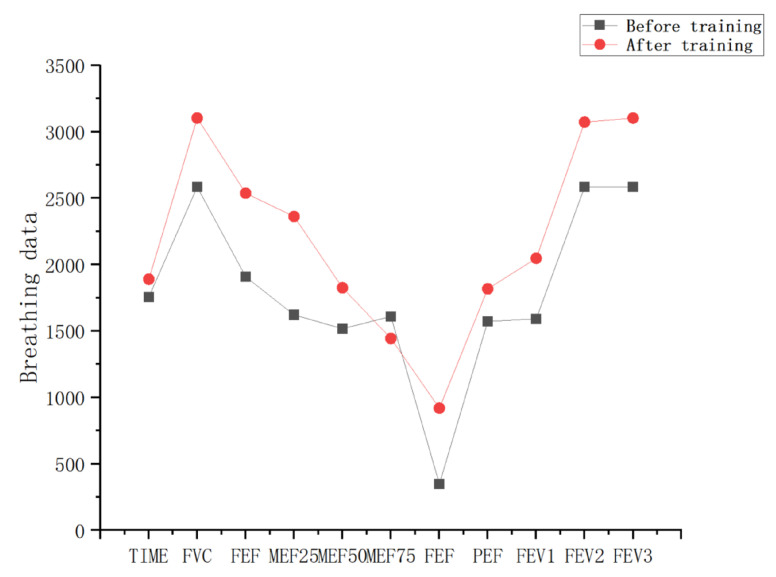
Average respiratory data of the observation group before and after training.

**Figure 12 sensors-23-09025-f012:**
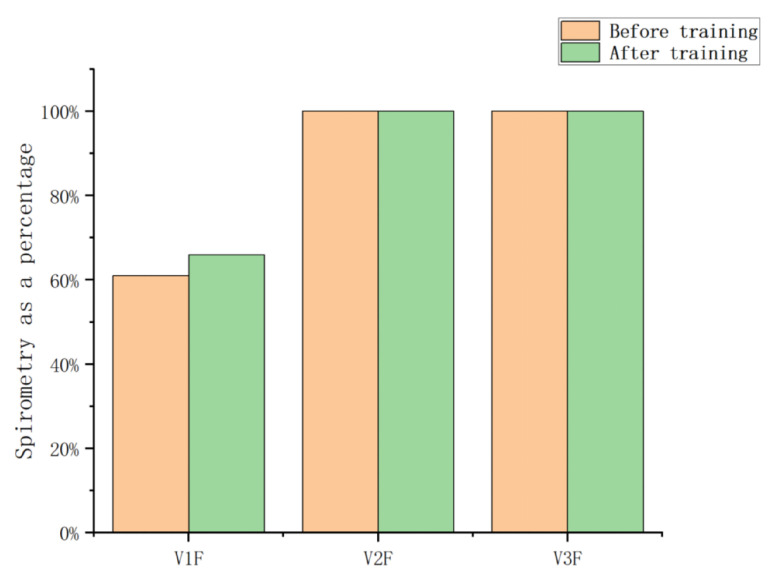
The average vital capacity ratio of the observation group before and after training.

**Figure 13 sensors-23-09025-f013:**
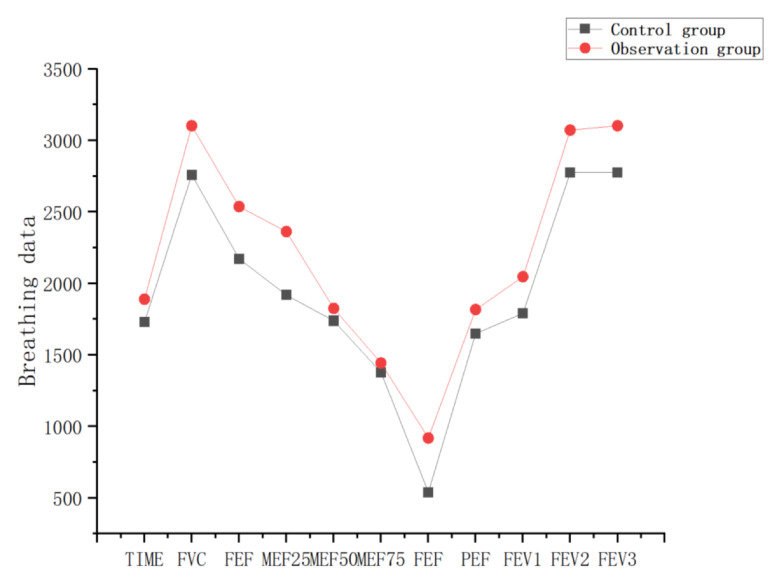
Respiration data before and after training of the control group and observation group.

**Figure 14 sensors-23-09025-f014:**
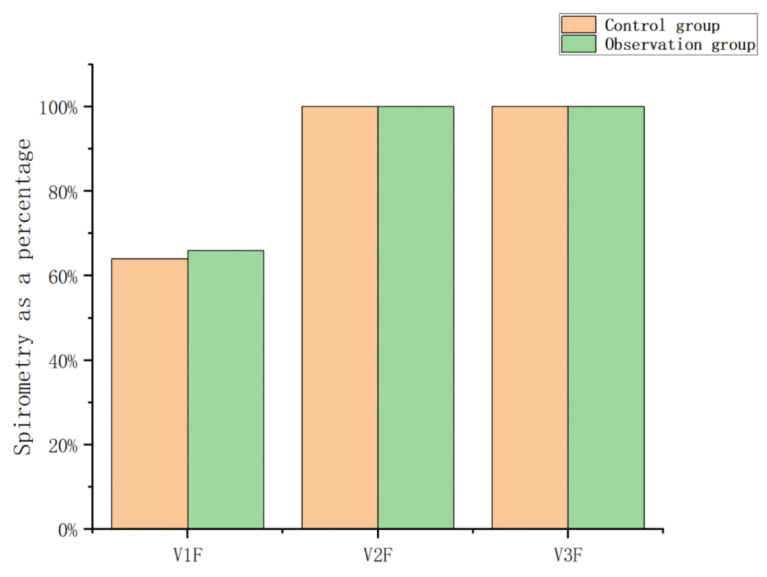
The ratio of lung capacity before and after training in the control group and the observation group.

**Figure 15 sensors-23-09025-f015:**
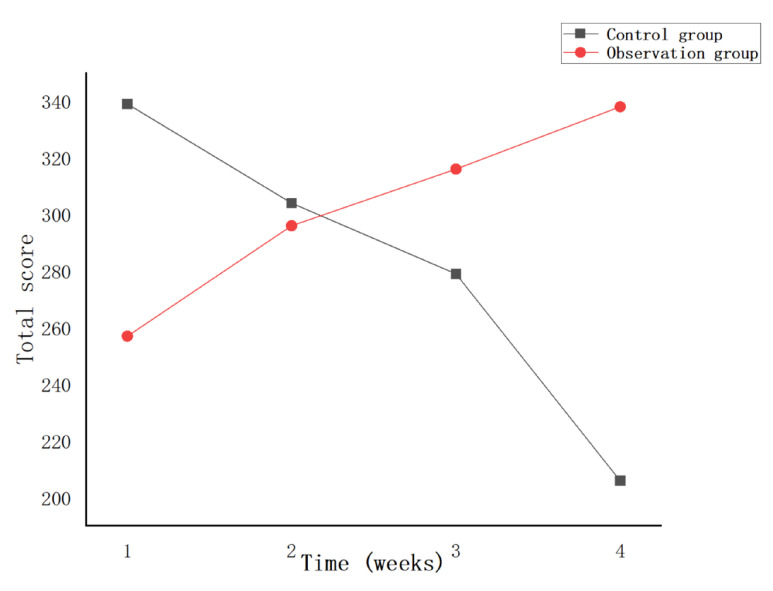
Total score of boredom per week.

**Figure 16 sensors-23-09025-f016:**
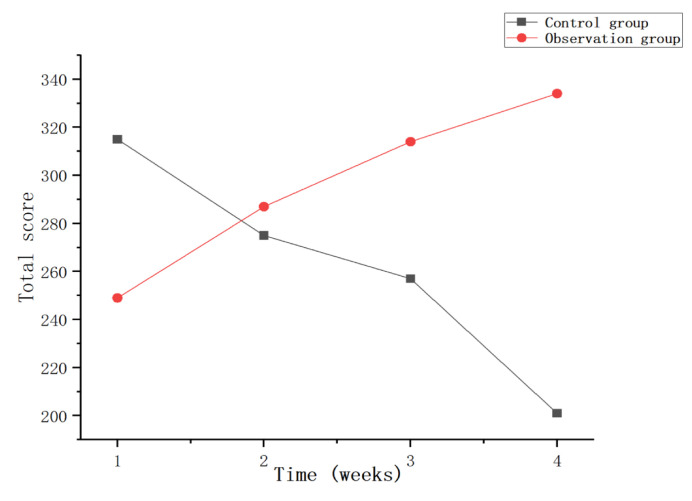
Total Weekly Inattention Scores.

**Figure 17 sensors-23-09025-f017:**
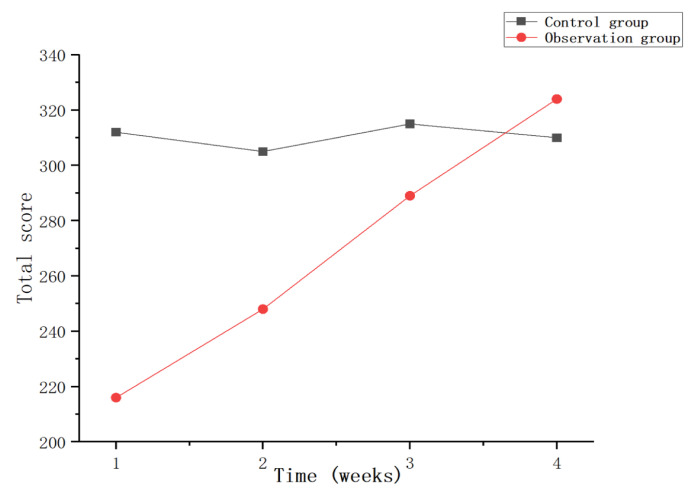
Total score for failing to keep up with the rhythm every week.

**Table 1 sensors-23-09025-t001:** Twenty-five respiratory data type bytes.

Byte Number	High and Low Byte Types	Respiration Data Type	
0	TIMEH Exhalation time high byte	Exhalation time	TIME
1	TIMEL Expiration time low byte		
2	FVCH Vital Capacity FVC High Byte	Forced vital capacity	FVC
3	FVCL Vital Capacity FVC Low Byte		
4	FEEH Peak flow rate high byte	Peak flow rate	FEF
5	FEEL Peak flow rate low byte		
6	MEF25H Flow rate high byte	Flow rate at 25% vital capacity	MEF25
7	MEF25L Flow rate low byte		
8	MEF50H Flow rate high byte	Flow rate at 50% vital capacity	MEF25
9	MEF50L Flow rate low byte		
10	MEF75H Flow rate high byte	Flow rate at 75% vital capacity	MEF25
11	MEF75L Flow rate low byte		
12	FEF25-75H 25–75% flow rate difference high byte	Exhalation time	FEF25-75
13	FEF25-75L 25–75% flow rate difference low byte		
14	PEF25-75H 25–75% average velocity high byte	Average flow rate	PEF25-75
15	PEF25-75L 25–75% average flow rate low byte		
16	FEV1H Vital capacity high byte in the previous second	Vital capacity in one second	FEV1
17	FEV1L Vital capacity low byte in the previous second		
18	FEV2H Vital capacity high byte in the first two seconds	Two-second vital capacity	FEV2
19	FEV2L Vital capacity low byte in the first two seconds		
20	FEV3H Vital capacity high byte in the first three seconds	Three-second vital capacity	FEV3
21	FEV3L Vital capacity low byte in the first three seconds		
22	V1F Vital capacity in one second as a percentage of FEV1/FVC	Unit:%	
23	V2F Vital capacity in two seconds as a percentage of FEV2/FVC		
24	V3F Three-second vital capacity as a percentage of FEV3/FVC		

**Table 2 sensors-23-09025-t002:** Participant information.

Participant ID	Gender	Age	Other Medical History	Tolerance	V1F
1	Man	60	NO	Well	<70%
2	Woman	63	NO	Well	<70%
3	Woman	64	NO	Well	<70%
4	Man	59	NO	Well	<70%
5	Man	57	NO	Well	<70%
6	Man	64	NO	Well	<70%
7	Man	65	NO	Well	<70%
8	Woman	56	NO	Well	<70%
9	Woman	62	NO	Well	<70%
10	Man	58	NO	Well	<70%

**Table 3 sensors-23-09025-t003:** Experimental Training Feeling Scale.

Serial Number	Question	Very Much Agree	Agree	General	Disagree	Strongly Disagree
1	During training, do you feel bored?	1	2	3	4	5
2	During training, are you distracted and unable to concentrate?	1	2	3	4	5
3	During training, are you unable to keep up with the training pace?	1	2	3	4	5

**Table 4 sensors-23-09025-t004:** The breathing data of the participants in the control group before training.

ID	TIME	FVC	FEF	MEF25	MEF50	MEF75	FEF	PEF	FEV1	FEV2	FEV3	V1F	V2F	V3F
	**(ms)**	**(mL)**	**(mL/s)**	**(mL/s)**	**(mL/s)**	**(mL/s)**	**(mL/s)**	**(mL/s)**	**(mL)**	**(mL)**	**(mL)**	**(%)**	**(%)**	**(%)**
1	1810	2311	1529	1297	1161	1487	190	1298	1244	2311	2311	53	100	100
2	1910	2497	1615	1601	1399	1001	600	1314	1494	2497	2497	59	100	100
3	1690	2689	1860	1729	1614	1605	124	1680	1700	2689	2689	63	100	100
4	1590	2420	1909	1667	1432	1274	393	1440	1585	2420	2420	65	100	100
5	1620	2431	1883	1747	1578	1757	290	1599	1668	2431	2431	68	100	100
AVG	1724	2470	1760	1609	1437	1425	319	1466	1538	2470	2470	62	100	100

**Table 5 sensors-23-09025-t005:** Breathing data of participants in the control group after training.

ID	TIME	FVC	FEF	MEF25	MEF50	MEF75	FEF	PEF	FEV1	FEV2	FEV3	V1F	V2F	V3F
	**(ms)**	**(mL)**	**(mL/s)**	**(mL/s)**	**(mL/s)**	**(mL/s)**	**(mL/s)**	**(mL/s)**	**(mL)**	**(mL)**	**(mL)**	**(%)**	**(%)**	**(%)**
1	1870	2719	1980	1652	1372	1312	340	1373	1555	2791	2791	57	100	100
2	1790	2686	2247	1825	1599	1244	581	1444	1688	2686	2686	62	100	100
3	1730	2582	1862	1712	1544	1355	357	1555	1649	2582	2582	63	100	100
4	1620	3379	2653	2396	2537	1646	750	2252	2341	3379	3379	69	100	100
5	1640	2432	2118	2010	1642	1335	675	1621	1724	2432	2432	70	100	100
AVG	1730	2760	2172	1919	1739	1378	540	1649	1791	2774	2774	64	100	100

**Table 6 sensors-23-09025-t006:** Respiration data of participants in the observation group before training.

ID	TIME	FVC	FEF	MEF25	MEF50	MEF75	FEF	PEF	FEV1	FEV2	FEV3	V1F	V2F	V3F
	**(ms)**	**(mL)**	**(mL/s)**	**(mL/s)**	**(mL/s)**	**(mL/s)**	**(mL/s)**	**(mL/s)**	**(mL)**	**(mL)**	**(mL)**	**(%)**	**(%)**	**(%)**
6	1640	2543	1771	1708	1614	1489	219	1609	1665	2543	2543	65	100	100
7	1970	2393	1544	1410	1223	1078	332	1208	1360	2393	2393	56	100	100
8	1590	2908	2634	1941	1710	2634	639	1964	1870	2908	2908	64	100	100
9	1740	2833	1975	1539	1901	1716	177	1839	1685	2833	2833	59	100	100
10	1840	2244	1614	1509	1138	1129	380	1246	1381	2244	2244	61	100	100
AVG	1756	2584	1908	1621	1517	1609	349	1573	1592	2584	2584	61	100	100

**Table 7 sensors-23-09025-t007:** Respiration data of participants in the observation group after training.

ID	TIME	FVC	FEF	MEF25	MEF50	MEF75	FEF	PEF	FEV1	FEV2	FEV3	V1F	V2F	V3F
	**(ms)**	**(mL)**	**(mL/s)**	**(mL/s)**	**(mL/s)**	**(mL/s)**	**(mL/s)**	**(mL/s)**	**(mL)**	**(mL)**	**(mL)**	**(%)**	**(%)**	**(%)**
6	1590	2777	2327	2291	1785	1735	556	1851	1957	2777	2777	70	100	100
7	2180	3471	2620	2461	1751	1287	1174	1770	2117	3418	3471	60	100	100
8	1820	2903	2226	2062	1786	1418	644	1748	1918	2903	2903	66	100	100
9	1990	3160	2497	2134	1779	1396	738	1736	1990	3160	3160	62	100	100
10	1870	3205	3013	2864	2025	1382	1482	1978	2251	3205	3205	70	100	100
AVG	1890	3103	2537	2362	1825	1444	919	1817	2047	3072	3103	66	100	100

## Data Availability

Not applicable.

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
