# Peer review of "Biofeedback Respiratory Rehabilitation Training System Based on Virtual Reality Technology"

_sensors, 2023, doi:10.3390/s23229025_

Round 1

Reviewer 1 Report

Comments and Suggestions for Authors

Thank you for the invitation to review this article.

It is important, especially when a new technology is presented, to do it with a larger number of subjects. In this case there are few subjects and it is not known how the randomisation was done and how they were diagnosed.

I recommend a sample size study to determine the number of subjects for the start of the trial.

The results are understood to be in ml but are not shown in the tables.

The discussion and conclusion should be modified according to the new sample size.

Author Response

Revision Summary

Manuscript ID:  sensors-2638328

Title: Biofeedback Respiratory Rehabilitation Training System Based on Virtual Reality Technology

We would like to thank the reviewers for their time and invaluable comments. We have carefully addressed all of their comments. Our detailed corresponding changes and other refinements we made are summarized as follows. Note that only these comments that suggest clarification, improvement or correction are quoted and responded below. Other positive comments are not repeated here but kindly appreciated.

Responses to the reviewer’s comments:

1) The reviewer states: “It is important, especially when a new technology is presented, to do it with a larger number of subjects. In this case there are few subjects and it is not known how the randomisation was done and how they were diagnosed.”

Response: Thanks to the reviewers for their comments. We apologize for the unclear description of how participants will be screened and diagnosed. This suggestion will help improve the academic rigor of the article. Based on the reviewers' comments, the article explains the basis for the selection of the 10 participants and their typical characteristics. To protect the privacy of participants’ personal information, Table 6 provides a brief introduction to the participants’ information. See revised version on page 11 for details.

2) The reviewer states: “The results are understood to be in ml but are not shown in the tables.”

Response: Thanks to the reviewers for their comments. This suggestion helps improve the readability of this study. As suggested by the reviewers, the respiratory data parameter units have been rewritten in the revised version. See revisions on pages 13 and 14 for details.

Reviewer 2 Report

Comments and Suggestions for Authors

The manuscript decribes a rehabilitation training system utilizing virtual reality and targeted on patients with respiratory problems. I understand the study as a feasibility study that had as its main objective to prove that the described approach can enhance the respiration rehabilitation.

The system was tested on 10 subjects. That means that it proved the feasibility. However, results from 10 patients cannot be so easily generalized as it was formulated in the manuscript. That is one of the points that should be re-formulated.

Moreover, there is no additional information about the subjects - demographic data, patient history, etc. showing homogenity/heterogeneity of the tested groups. 

The comparison of results reached with and without VR is well performed.

I have also a question concerning the training. From the description it follows that only two different scenarios/scenes (candle and dandelion) were displayed to the tested subjects. Do the authors think that in case more options (more scenes) were offered to the subjects it would result in increased motivation to train (i.e. higher user acceptance)?

Section 4.2 - it is not necessary to explain what is the Lkert scale. A reference to the original publication is satisfactory.

Comments on the Quality of English Language

There are some minor issues in formulations. I recommend a check by a native speaker.

Author Response

Revision Summary

Manuscript ID:  sensors-2638328

Title: Biofeedback Respiratory Rehabilitation Training System Based on Virtual Reality Technology

We would like to thank the reviewers for their time and invaluable comments. We have carefully addressed all of their comments. Our detailed corresponding changes and other refinements we made are summarized as follows. Note that only these comments that suggest clarification, improvement or correction are quoted and responded below. Other positive comments are not repeated here but kindly appreciated.

Responses to the reviewer’s comments:

1) The reviewer states: “The system was tested on 10 subjects. That means that it proved the feasibility. However, results from 10 patients cannot be so easily generalized as it was formulated in the manuscript. That is one of the points that should be re-formulated. Moreover, there is no additional information about the subjects - demographic data, patient history, etc. showing homogenity/heterogeneity of the tested groups. ”

Response: Thanks to the reviewers for their comments. This suggestion will help improve the academic rigor of the article. Based on the reviewers' comments, the article explains the basis for the selection of the 10 participants and their typical characteristics. To protect the privacy of participants’ personal information, Table 6 provides a brief introduction to the participants’ information. See revised version on page 11 for details.

2) The reviewer states: “I have also a question concerning the training. From the description it follows that only two different scenarios/scenes (candle and dandelion) were displayed to the tested subjects. Do the authors think that in case more options (more scenes) were offered to the subjects it would result in increased motivation to train (i.e. higher user acceptance)?”

Response: Thank you for the questions raised by the reviewer. Let us answer your question: It is not that providing more training scenarios for students will lead to better training results, but that different training scenarios are suitable for different training groups, and the training levels of different training scenarios are also different.

3) The reviewer states: “Section 4.2 - it is not necessary to explain what is the Lkert scale. A reference to the original publication is satisfactory.”

Response: Thanks for the reviewers’ comment. This suggestion is very valuable for revising the manuscript. We have made modifications to the introduction of the scale. Please see the revision for details on page 11.

Reviewer 3 Report

Comments and Suggestions for Authors

Without any doubt a very interesting research topic: designing and prototyping a respiratory rehabilitation training system based on virtual reality.

This research topic appeared really interesting.

I am not sure at al this system may be classified as virtual reality system... or simply computer aided respiratory rehabilitation system. Patient is interacting with a software that appears more or less such as a game.

My only concern is the article content is a bit too descriptive and many event types are not sufficiently presented/discussed.

For example, data V1F, v2F and V3F are not really well introduced. These data appear really important to evaluate the efficiency of the proposed application.

For example, it is not so evident the analyse of these data (Figure 10) may allow to conclude of the effectiveness of the proposed solution. It is too descriptive. Please explain deeper.

I think you have to modify the Chinese writing within Figures 5 and 8, and then provide English written ones.

Well polished this article might be good.

Author Response

Revision Summary

Manuscript ID:  sensors-2638328

Title: Biofeedback Respiratory Rehabilitation Training System Based on Virtual Reality Technology

We would like to thank the reviewers for their time and invaluable comments. We have carefully addressed all of their comments. Our detailed corresponding changes and other refinements we made are summarized as follows. Note that only these comments that suggest clarification, improvement or correction are quoted and responded below. Other positive comments are not repeated here but kindly appreciated.

Responses to the reviewer’s comments:

1) The reviewer states: “data V1F, v2F and V3F are not really well introduced.”

Response: Thanks for the reviewer’s comment. This suggestion is very valuable for revising the manuscript. In the text, we will add descriptions of V1F, v2F,and V3F:It is observed in Tabl6 and Table7that the vital capacity in the first second of all patients is low, but the vital capacity in the second and third seconds is high. Because diseases such as pulmonary obstruction, emphysema, asthma, etc. can cause airway obstruction, making the lungs unable to exhale the full lung capacity in the first second. However, over time, lung capacity gradually increases in the second and third seconds, and the patient tries to expel as much gas as possible by increasing their breathing rate or changing their breathing pattern. Please see the revision for details on page 12.

2) The reviewer states: “I think you have to modify the Chinese writing within Figures 5 and 8, and then provide English written ones.”

Response: Thank you for the reviewer's comments. This suggestion can contribute to promote the readability of this study. According to the reviewer's suggestions, Figures 5 and 8 have been modified in this article.

Reviewer 4 Report

Comments and Suggestions for Authors

The paper proposes a biofeedback respiratory rehabilitation training system based on virtual reality technology by collecting and preprocessing respiratory data through a respiratory sensor. At the same time, it combines the biofeedback respiratory rehabilitation training virtual scene to realize the interaction between respiratory data and virtual sets.

The paper is well organized, and the length is appropriate. The title is chosen correctly, and the abstract provides sufficient information to understand what to expect from the paper.

The study meets ethical requirements, and informed consent was obtained from all subjects involved.

The introduction is well structured and covers almost all the concepts investigated in the methodological part.

The research design used is appropriate to answer the research questions proposed by the authors. The methods are correctly described. The results are clearly presented and are in relation to the concepts investigated. However, the limitations of the study should be underlined.

The conclusions are strongly related to the findings of the research work.

The references are relevant and correctly chosen, and related work is discussed and cited appropriately. However, other articles could also be considered.

Despite the good work done, there is still some room for improvement, as follows:

1.      You wrote, in row 19, “… in my country has exceeded 100 million…” maybe it is better to use “.. in our country…” or to use an impersonal style such as: “In China, the number of patients with chronic obstructive pulmonary disease (COPD) has exceeded 100 million, and is still on the rise, according to the "China Adult Lung Health Study" released in 2022.”

2.      In the introduction or related work section, some studies must be included where solutions on using virtual reality in general rehabilitation and for COPD, in particular, are presented. As a suggestion could be the following papers:

-        Patsaki, I., Avgeri, V., Rigoulia, T., Zekis, T., Koumantakis, G.A., Grammatopoulou, E., 2023. Benefits from Incorporating Virtual Reality in Pulmonary Rehabilitation of COPD Patients: A Systematic Review and Meta-Analysis. Advances in Respiratory Medicine 91, 324–336.. https://doi.org/10.3390/arm91040026

-        Ciorap R., AndriÅ£oi D., Casuţă A., Ciorap M., Munteanu D., "Game-based virtual reality solution for post-stroke balance rehabilitation." IOP Conference Series: Materials Science and Engineering, vol. 1254, no. 1, 2022, pp. 012037. DOI: 10.1088/1757-899x/1254/1/012037

-        Liu, W.-Y., Meijer, K., Delbressine, J., Willems, P., Wouters, E., Spruit, M., 2019. Effects of Pulmonary Rehabilitation on Gait Characteristics in Patients with COPD. Journal of Clinical Medicine 8, 459.. https://doi.org/10.3390/jcm8040459

-        Rutkowski, S.; Buekers, J.; Rutkowska, A.; CieÅ›lik, B.; Szczegielniak, J. Monitoring Physical Activity with a Wearable Sensor in Patients with COPD during In-Hospital Pulmonary Rehabilitation Program: A Pilot Study. Sensors 2021, 21, 2742. https://doi.org/10.3390/s21082742

3.      In the conclusion section, the limitations of the study should be underlined.

4.      In rows 38-39, you wrote “ NOMAN SADIQ[9]” with uppercase.

5.      In row 200, you wrote: ” he source of respiratory data is collected…”  probably was “The source…” please correct. 

Author Response

Manuscript ID:  sensors-2638328

Title: Biofeedback Respiratory Rehabilitation Training System Based on Virtual Reality Technology

We would like to thank the reviewers for their time and invaluable comments. We have carefully addressed all of their comments. Our detailed corresponding changes and other refinements we made are summarized as follows. Note that only these comments that suggest clarification, improvement or correction are quoted and responded below. Other positive comments are not repeated here but kindly appreciated.

Responses to the reviewer’s comments:

1) The reviewer states: “You wrote, in row 19, “… in my country has exceeded 100 million…” maybe it is better to use “.. in our country…” or to use an impersonal style such as: “In China, the number of patients with chronic obstructive pulmonary disease (COPD) has exceeded 100 million, and is still on the rise, according to the "China Adult Lung Health Study" released in 2022.””

Response: Thanks for the reviewer’s comment. This suggestion helps to improve the quality of writing this article. In the text, we modify it to:In China, the number of patients with chronic obstructive pulmonary disease (COPD) has exceeded 100 million, and is still on the rise, according to the "China Adult Lung Health Study" released in 2022.

2) The reviewer states: “In the introduction or related work section, some studies must be included where solutions on using virtual reality in general rehabilitation and for COPD, in particular, are presented.”

Response: Thanks to the reviewers for their comments. This suggestion will help improve the academic rigor of the article. Based on the reviewers’ comments, the revised manuscript provides relevant reference requirements [4, 6, 20, 22] for solutions using virtual reality in rehabilitation and solutions for chronic obstructive pulmonary disease.

3) The reviewer states: “ In the conclusion section, the limitations of the study should be underlined.”

Response: Thank you for the reviewer's comments. This suggestion helps to illustrate the next direction of this study. According to the reviewer's suggestion, the following has been added in the conclusion section:Due to the interference of respiratory equipment in the process of respiratory data transmission, certain Gaussian white noise and random noise will be generated. The focus of follow-up research is to further improve and perfect the respiratory data collection and improve the accuracy of respiratory data transmission. Please see the revision for details on page 18.

4) The reviewer states: “ In rows 38-39, you wrote “ NOMAN SADIQ[9]” with uppercase.”

Response: Thank you for your comments. This suggestion helps to improve the readability of this study. Based on the reviewer's suggestions, I have made modifications in this article:Noman Sadiq[9]. Please see the revision for details on page 2.

5) The reviewer states: “ In row 200, you wrote: ” he source of respiratory data is collected…”  probably was “The source…” please correct.”

Response: Thank you for your comments. This suggestion is conducive to improve the academic rigor of the article. Based on the reviewer's suggestions, I have made modifications in this article:The respiratory data is collected through the sensor, and the data signal output of the vital capacity sensor is connected to the computer through the serial port, and the data is transmitted to the PC in the form of serial communication. Please see the revision for details on page 7.

Round 2

Reviewer 1 Report

Comments and Suggestions for Authors

According to the changes made

Reviewer 2 Report

Comments and Suggestions for Authors

The manuscript has been improved. The authors responded satisfactorily in the text to the reviewer´s comments.